# The Protein Response of Salt-Tolerant *Zygosaccharomyces rouxii* to High-Temperature Stress during the Lag Phase

**DOI:** 10.3390/jof10010048

**Published:** 2024-01-05

**Authors:** Na Hu, Xiong Xiao, Lan Yao, Xiong Chen, Xin Li

**Affiliations:** Key Laboratory of Fermentation Engineering (Ministry of Education), Cooperative Innovation Center of Industrial Fermentation (Ministry of Education & Hubei Province), Hubei Key Laboratory of Industrial Microbiology, School of Biological Engineering and Food, Hubei University of Technology, Wuhan 430068, China; 102110674@hbut.edu.cn (N.H.); 102200529@hbut.edu.cn (X.X.); yaolislan1982@aliyun.com (L.Y.)

**Keywords:** *Zygosaccharomyces rouxii*, high temperature, ubiquitin–proteasome system, amino acid biosynthesis, amino acid composition of proteins

## Abstract

*Zygosaccharomyces rouxii* used in soy sauce brewing is an osmotolerant and halotolerant yeast, but it is not tolerant to high temperatures and the underlying mechanisms remain poorly understood. Using a synthetic medium containing only Pro as a nitrogen source, the response of *Z. rouxii* in protein level to high-temperature stress (40 °C, HTS) during the lag phase was investigated. Within the first two h, the total intracellular protein concentration was significantly decreased from 220.99 ± 6.58 μg/mg DCW to 152.63 ± 10.49 μg/mg DCW. The analysis of the amino acid composition of the total protein through vacuum proteolysis technology and HPLC showed that new amino acids (Thr, Tyr, Ser, and His) were added to newborn protein over time during the lag phase under HTS. The nutritional conditions used in this study determined that the main source of amino acid supply for protein synthesis was through amino acid biosynthesis and ubiquitination-mediated protein degradation. Differential expression analysis of the amino acid biosynthesis-related genes in the transcriptome showed that most genes were upregulated under HTS, excluding ARO8, which was consistently repressed during the lag phase. RT-qPCR results showed that high-temperature stress significantly increased the upregulation of proteolysis genes, especially *PSH1* (E3 ubiquitin ligase) by 13.23 ± 1.44 fold (*p* < 0.0001) within 4 h. Overall, these results indicated that *Z. rouxii* adapt to prolonged high temperatures stress by altering its basal protein composition. This protein renewal was related to the regulation of proteolysis and the biosynthesis of amino acids.

## 1. Introduction

Microorganisms are highly sensitive to temperature changes, which can significantly impact the structure and function of their proteins. High-temperature stress can lead to protein misfolding and aggregation, which in eukaryotic cells can trigger the formation of proteins and RNA into stress granules [1]. Proteins play a vital role in promoting yeast growth and can withstand the stress response by breaking down or producing new proteins. Moreover, by modifying their basic components, proteins can respond physiologically to sustained high temperatures. Yeast can increase the thermodynamic stability of highly expressed proteins by expanding the range of amino acid substitutions that proteins can withstand at high temperatures, thus reducing the burden of intracellular protein misfolding [2]. Substitutions in the main amino acid composition of proteins can impede amino acid biosynthesis pathways. For instance, Thakre et al. [3] demonstrated that substituting Arg, a major amino acid component of histone residues, with Ala made *Saccharomyces cerevisiae* growth dependent on Ile, Ser, or Thr supplementation, indicating that Ile biosynthesis was hindered.

Maintenance of the proteome is important in the stress response system [4]. This dynamic equilibrium is mediated by the coupling of ubiquitin with intracellular proteins for their selective degradation in eukaryotes. Among the primary pathways on which proteostasis relies is the ubiquitin–proteasome system. Under stress conditions, such as high temperature, cells must rapidly adapt by increasing their folding capacity to eliminate abnormal proteins. Proteostasis in this way requires a balance of protein synthesis, folding, and degradation [5]. In *S. cerevisiae*, Benet et al. [6] observed that cellular gene translation capacity and protein degradation rates increased in response to changes in external temperature. Mühlhofer et al. [7] further explained that 90% of heat-induced gene upregulation was due to the yeast’s need to maintain protein homeostasis by directly balancing increased stress-dependent protein aggregation and degradation processes. This protein regulation is not solely a product of the stress response but may also come from active selection by the yeast itself to achieve an optimal mode of adaptation. For instance, Will et al. [8] found that the loss of aquaporins (Aqy1 and Aqy2) in the *S. cerevisiae* strain grown at high sugar concentrations was due to positive active selection rather than suppression of expression.

The physiological mechanisms of *Z. rouxii* enduring salt and sugar stress in a variety of foods have been widely investigated. Guo et al. [9] explained that *Z. rouxii* acts to cope with high sugar tolerance at the protein molecular level in concentrated juice. High-temperature stress is a condition that commonly affects yeast viability in most brewing systems. However, few studies have investigated the protein level of *Z. rouxii* at high temperatures. This paper proposed an analysis of two pathways, the biosynthesis of amino acids and the ubiquitin–proteasome system, in investigating the regulation of *Z. rouxii* under high-temperature stress (HTS). Meanwhile, lag is a dynamic, organized, adaptive, and evolvable process that protects bacteria from threats, promotes reproductive fitness, and is broadly relevant to the study of bacterial evolution, host-pathogen interactions, antibiotic tolerance, environmental biology, molecular microbiology, and food safety [10]. However, this was still a small number of studies, with the majority focusing on changes in the stress response of log phase yeast. Because of the short duration of the lag phase and the unclear composition of the medium, it was difficult to obtain valid data. Therefore, in this paper, we created a synthetic medium in which yeast cell was grown in minimal nutrition condition. The standard temperature for normal physiologic growth of yeast is 30 °C, while 40 °C is the temperature used to induce high-temperature stress. By comparing these two representative temperature conditions, the physiological and transcriptional levels of *Z. rouxii* in a minimal synthetic medium were determined.

## 2. Materials and Methods

### 2.1. Measurement of Yeast Cultivation and Growth Curve

The *Z. rouxii* M2013310 strain (CCTCC M 2013310) isolated from the traditional Chinese food brewing was used. The strain was streaked onto YPD agar from a −80 °C glycerol stock (strain/glycerol = 1/1, *v*/*v*), followed by incubation at 30 °C for 48 h. In general, Pro was considered to be an important amino acid for *S. cerevisiae* in the face of stress response [11]. Therefore, the growth pattern of *Z. rouxii* was observed in minimal synthetic medium (50 mL) (glucose, 4.0 g/L L-proline, 1.0 mg/L pantothenate, 1.0 mg/L folic acid, 1.0 g/L KH_2_PO_4_, and 0.5 g/L MgSO_4_) under dynamic conditions (initial cell density OD_600_ = 0.5 and 200 RPM) for 24 h. Cells were grown under two conditions, namely control (30 °C, glucose 20 g/L) and high-temperature stress (40 °C, glucose 300 g/L) (HTS). The number of viable cells was monitored every 2 h using plate count agar. The cell density of yeast at 120 h under HTS was determined in the evaluation test of amino acid growth promotion in which Pro was replaced with Lys, Tyr, or Phe in equal amounts. The yeast cell culture optical density (OD_600_) was monitored with a visible spectrophotometer V-1300 (Macylab, Sanghai, China). All experiments were carried out using three biological replicates.

### 2.2. Analysis of Amino Acid of the Total Protein (AA-P)

Methods for collecting the cells and protein extraction, identification, and quantification were described in Zaman et al. [12]. A SCIENTZ-II ultrasonic processor (130 W, 3–4 s intervals, 20 min) was used to disrupt the cells. Protein concentration was determined using a BCA protein assay kit (Biosharp, Beijing, China) and analyzed by Nanodrop 2000 (Thermo Fisher Scientific, Vantaa, Finland).

The protein liquid was collected in a 20 mL penicillin vial and immediately dried using a vacuum freeze dryer (Labconco FreeZone, Thermo Fisher Scientific, Vantaa, Finland) for 24 h. The dried sample was then transferred to a 20 mL hydrolysis tube, and 5 mL of 6 mol/L HCl solution was added. The tube was degassed under vacuum for 30 min, sealed with nitrogen, and then hydrolyzed at 110 °C for 24 h. After cooling, the volume was fixed to 5 mL with ddH_2_O. Then, a 1.0 mL sample of the hydrolysis solution was deacidified under vacuum and drained. This process was repeated with 1.0 mL of water added each time. Finally, 1.0 mL of 0.02 mol/L HCl was added to thoroughly dissolve the sample, followed by 250 μL of 1.0 mol/L triethylamine acetonitrile solution. The solution was shaken well to form the amino acid solution.

### 2.3. Detection of Free Amino Acid (AA-F)

Methods for cell collection and grinding were described in Canelas et al. [13]. Cells were quenched using 2 mL of 400 g/L ethanol and 0.5 mm glass beads (cells/beads: 1/5, mg/mg). The supernatant was extracted and combined with an isolated culture fluid. The 500 µL liquid was added to 45 µL of 7 mol/L NaOH solution and then 500 µL of a mixture of anhydrous ethanol and pyridine (*v*/*v*: 4/1) for amino acid extraction were added to make an amino acid solution.

### 2.4. Determination of Amino Acid Composition

The amino acid solutions were derivatized according to the method described by Klikarova [14] and analyzed by ultra-performance liquid chromatography with a UV detector (U3000, Thermo Fisher, MA, USA) and a C18 column (4.6 mm × 250 mm, 5 μm). The temperature was set at 40 °C, and detection was performed at 254 nm. The eluents used were (A) acetonitrile–water (80:20, *v*/*v*) and (B) diH_2_O /sodium acetate buffer solution (pH 6.5)–acetonitrile (93:7, *v*/*v*) at a flow rate of 1.0 mL/min. The following gradient was used (*v*/*v*): 0.0 min, 0% A; 11.0 min, 7% A; 12.0 min, 12% A; 14.0 min, 15% A; 29.0 min, 34% A; 32.0 min, 70% A; 35.0 min, 100% A; and 45.0 min, 0% A. Amino acid composition calculation method was the average of the ratio of the individual amino acid content to the total measured amino acid content in this culture system.

### 2.5. Eukaryotic Transcriptome Sequencing under HTS

#### 2.5.1. Extraction and Quality Assessment of Total RNA

Yeast pellets (≥100 mg) were collected and immediately washed with 4 °C DEPC (diethyl pyrocarbonate) water (Biosharp, Beijing, China) and repeated three times by centrifugation for 5 min at 13,400× *g* at 4 °C. Total RNA was extracted from yeast cells using the TRIzol^®^ kit (Thermo Fisher Scientific, Vantaa, Finland), and genomic DNA was removed using DNase I (Thermo Fisher Scientific, Vantaa, Finland). RNA purity and concentration were measured using the NanoDrop2000 RNA assay (Thermo Fisher Scientific, Vantaa, Finland), and RNA integrity number (RIN) was measured using the Agilent 2100 Nano (Agilent, CA, USA). Three biological replicates were performed.

#### 2.5.2. Library Preparation and Sequencing

RNA purification, reverse transcription, library construction, and sequencing were performed according to the manufacturer’s instructions at Shanghai Biomajorbio Bio-Pharm Technology Company. RNA-seq transcriptome libraries were constructed using the Illumina TruSeqTM RNA Sample Prep Kit [15]. PCR amplification was then performed using Phusion DNA polymerase for 15 PCR cycles. After quantification by TBS380, the double-ended RNA-seq sequencing library was sequenced onboard using an Illumina NovaSeq 6000 (URL www.illumina.com.cn) (2 × 150 bp read length). RNA-Seq data were uploaded to the NCBI Sequence Read Archive with the registration number (PRJNA975303).

#### 2.5.3. Transcriptome Data Analysis

Raw end reads were cropped and quality controlled using the software SeqPrep 1.2 (https://github.com/jstjohn/SeqPrep accessed on 10 May 2023) and Sickle 1.1.3 (https://github.com/najosh-i/sickle accessed on 10 May 2023) with default parameters [16]. Clean reads were individually compared to the reference genome using the software TopHat 8.48.0 (http://tophat.cbcb.umd.edu/, version 2.1.1 accessed on 10 May 2023) in directed mode [17]. Differential expression analysis was performed using the DESeq2 1.42.0 (http://bioconductor.org/packages/stats/bioc/DESeq2 accessed on 15 May 2023) library for the statistical software R 3.6.2. Only genes with a false discovery rate (FDR) adjusted *p*-values < 0.05, and a log2 fold change (FC) > 2 were considered as differentially expressed. The information was then compared with the original genomic *Z. rouxii* GCA_001972345.1 (https://www.ncbi.nlm.nih.gov/assembly/GCA_001972345.1 accessed on 24 May 2023) KEGG annotation to screen for genes with differential expression in amino acid biosynthesis pathways and the ubiquitin-proteasome system. Three biological repeats per time point.

### 2.6. RT-qPCR Analysis

The raw data were compared with the *S. cerevisiae* (YDR316W) using Hisat2.2.1 PE reads survival after the trimming step > 90.00% and percentage of mapped reads > 90.00%. Eight validated genes were screened against references on *S. cerevisiae* proteolysis genes, amino acid biosynthesis pathway, and the KEGG pathway database in combination with transcriptome differential analysis and subjected to RT-qPCR validation. Eight differentially expressed genes in the ubiquitin-proteasome system and *ARO8* were selected. Total RNA was extracted from the simple total RNA kit (TIANGEN). RHiScript II Q RT SuperMix for qPCR (+gDNA wiper) (Vazyme, Nanjing, China) was utilized to convert RNA into cDNA on a T100 thermal cycler (Bio-Rad, Hercules, CA, USA). RT-qPCR with ChamQ Universal SYBR qPCR master mix (Vazyme, Nanjing, China) on selected genes was performed on a CFX96 Real-time PCR Detection System (Bio-Rad). Primer design (Table 1) was performed using Primer Premier 6.0. Three technological repeats were performed for each biological replicate. All kits were operated according to the manufacturer’s instructions. The obtained Cq values were used for further data analysis. Delta cycle threshold (ΔC_T_) values were calculated by the C_T_s of the target genes minus the C_T_ of *ENO1* [18], which was used as a housekeeping gene. The ΔΔC_T_ values waere calculated from the ratio of the ΔC_T_ values of the experimental sample to the C_T_ value of the control sample. Fold changes were calculated using the 2^−ΔΔCT^ method [19].

## 3. Results

### 3.1. The Effect of HTS on the Nutritional Requirements of Yeast Cells

According to the results proposed in Figure 1, *Z. rouxii* was found to grow effectively at 30 °C in both 20 g/L glucose and 300 g/L glucose in a minimal synthetic medium. At 40 °C, 20 g/L glucose only maintained cell survival, and 300 g/L glucose enabled effective growth conditions only at high temperatures in *Z. rouxii*. Based on the growth curves, it was observed that the growth progress of yeast under both 30 °C 20 g/L glucose and 40 °C 300 g/L glucose conditions overlapped at 14 h. However, cells at 30 °C 300 g/L glucose condition and 40 °C 20 g/L glucose condition cannot be assessed for the lag phase. It is generally accepted that the C/N ratio of 16.39 g C/g N (20 g/L glucose) was a medium with good nitrogen content to support good yeast growth, so we chose 30 °C 20 g/L glucose was selected as the control. Meanwhile, we observed that *Z. rouxii* grows better in a medium with 300 g/L glucose at a C/N ratio of 245.9 g C/g N, and this is especially evident at high temperatures. However, in most cases, we believe that a C/N ratio of 245.9 g C/g N with a worse nitrogen source should be detrimental to growth, so we further compared the fixed and free amino acids of these two conditions to understand the response mechanism of *Z. rouxii* under high temperature and high osmotic stress.

### 3.2. Total Protein Content and Its Amino Acid (AA-P) Composition under Two Conditions

Comparison of the trend of the total protein content of *Z. rouxii* (Figure 2A) showed that a significant difference (*p* < 0.0001) between the control and HTS appeared after 8 h. However, it was noteworthy that the total protein content in HTS during the lag phase (0 h to 4 h) showed the same trend as the control *(p* > 0.05), which was a decrease followed by an increase. For a more detailed analysis of the total protein response, the amino acid (AA-P) composition of the total protein was analyzed under two conditions of the lag phase (0 h to 4 h). The change in the amino acid composition of the *Z. rouxii* total protein was analyzed using the initial value (0 h) as a reference point for subsequent time points, which was the total content of AA-P was 2510.12 ± 261.03 mg/kg. In the AA-P composition under initial value (Figure 2B), 14 amino acids were detected, and the 5 main amino acids were Arg (27.49%), Leu (22.27%), Glu (8.44%), Gln (5.70%), and Asp (5.48%). The total amino acid content of the protein in the control was 1569.21 ± 230.17 mg/kg at 2 h and 1975.6 ± 141.50 mg/kg at 4 h. In the control, 14 amino acids were detected at 2 h (Figure 2C), but the amino acid composition changed, with the top five amino acids at 2 h being Glu (18.21%), Thr (13.69%), Lys (12.45%), Asp (7.79%), and Val (6.90%). Compared to the initial stage, three amino acids (Arg, Ala, and Hyp) disappeared, and three amino acids (Thr, Ser, and Tyr) increased at 2 h. The amino acid composition of the total protein at 4 h was the same as at 2 h, but the percentage of each amino acid changed.

The total amino acid content of the protein in the HTS was 1889.25 ± 174.09 mg/kg at 2 h and 3478.42 ± 309.99 mg/kg at 4 h. Under HTS (Figure 2D), 13 amino acids were detected at 2 h, and 17 amino acids were detected at 4 h. At 2 h, the top five amino acids were Arg (23.80%), Leu (15.65%), Glu (12.74%), Asp (10.18%), and Val (6.54%). Compared to the initial stage, two amino acids (Ala and Hyp) disappeared, and a new amino acid (Ser) was added. At 4 h, the top five amino acids were Arg (18.31%), Glu (14.30%), Asp (7.54%), Lys (7.19%), and Leu (6.30%). Four amino acids (Ala, Thr, Tyr, and His) were added at 4 h compared to 2 h. These results suggested that the amino acid composition of *Z. rouxii* proteins was strongly influenced by HTS. The cells constantly introduced new amino acids as they synthesized proteins under HTS. However, at normal physiological temperature (30 °C), AA-P was only an initial change, and no new amino acid components appeared afterward. The two conditions altered only the glucose content of the medium, suggesting that glucose compensated for the effects of HTS on *Z. rouxii*. Therefore, we need to further investigate the self-synthesis of amino acids by *Z. rouxii*.

### 3.3. Free Amino Acid (AA-F) Composition under Two Conditions

Pro was the only amino acid added to the synthetic medium, and its consumption can be used as a nitrogen source. According to the results (Figure 3A), both the HTS and the control were able to end the rapid proline depletion period in the first 12 h and there was no significant difference in the amount of Pro residue content (approximately 2800 mg/L). However, only at 2 h was there a significant difference (*p* < 0.001) in Pro content between the two conditions. We have therefore exhaustively analyzed the composition of free amino acids (AA-F) in addition to Pro during the lag phase (0 h to 4 h).

The total free amino acid content (except Pro) was 561.72 ± 22.63 mg/L at 2 h and 510.71 ± 24.99 mg/L at 4 h in the control, and 303.73 ± 25.13 mg/L at 2 h and 311.62 ± 27.77 mg/L at 4 h in the HTS. The composition of the free amino acid (except Pro) at 2 h and 4 h was analyzed (Figure 3B,C). Seven free amino acids, Asn, Met, Trp, Ser, Phe, Pro, and Glu, could be detected under the control (Figure 3B), in which the Asn (66.29% at 2 h and 61.54% at 4 h) continues to dominate role. Seven free amino acids (Trp, Asn, Met, Phe, Ile, Pro, and Glu) were detected under HTS (Figure 3C). At 2 h under HTS, the rank first amino acid was the same as that of the control, which was still Asn (36.23%). However, at 4 h under HTS, the rank first free amino acid was changed from Asn to Trp (30.54%). This phenomenon indicates that *Z. rouxii* accumulates a large amount of free Trp under HTS. This may imply that certain synthetic pathways were affected in *Z. rouxii* at HTS in the need to accumulate Trp to compensate for this deficiency, and we therefore proceeded to explore the transcriptional level of amino acid biosynthesis pathway genes under HTS.

### 3.4. Analysis from the KEGG Amino Acid Biosynthesis Pathway Map Combined with AA-F and AA-P under HTS

To probe the specific origin of amino acids, the use of the KEGG database is an effective means of validation. PATHWAY is a collection of manually created KEGG pathway maps representing molecular wiring diagrams of biological systems [20]. In the KEGG database for *Z. rouxii* (zro01230), a metabolic pathway was described that starts from glucose and details the biosynthesis of 21 amino acids (Figure 4A). The differential expression analysis results of 59 genes during the lag phase (0 h to 4 h) under HTS showed that most of the genes were upregulated, except for ARO8 (Figure 4B), which was clearly and consistently inhibited. *ARO8* was involved in the Tyr (zro_M00025), Lys (zro_M00030), and Phe (zro_M00024) biosynthesis pathways (Figure 5A–C). Furthermore, the downregulation of *ARO8* expression revealed on the transcriptome during HTS was confirmed by RT-qPCR analysis (Figure 5D). However, AA-F (Figure 3) still contained Phe, and there was approximately twice as much Phe as in the control under HTS. This was an indication that Phe biosynthesis was regulated post-transcriptionally under HTS.

Ser was not detected in AA-P (Figure 2B) at the initial stage (0 h). Under HTS, Ser was not detectable in AA-F (Figure 3C). However, the trend in the composition of AA-P (Figure 2D) under HTS showed that the proportion of Ser continued to increase (0%–3.52%–4.79%). Conversely, in the control, the percentage of Ser in AA-P (Figure 2C) increased (0%–3.93%–3.86%) but also in AA-F (Figure 3B). According to Figure 4A, Ser could not be biosynthesized directly from other amino acids. Further investigation of the KEGG amino acid biosynthesis pathway at the transcriptional level (Figure 4B) showed that Ser biosynthesis genes *SER1* and *SER2* (zro_M00020) were not inhibited under HTS. These results indicated that *Z. rouxii* needs to produce more Ser-containing proteins under HTS. In addition, we found that Trp could be biosynthesized by two precursors, namely Chorismate and Ser, and the transcriptional level was all significantly upregulated (*TRP1*, *TRP2*, *TRP3*, *TRP4,* and *TRP5*). After AA-F (Figure 3C), Trp gradually becomes the first free amino acid. We further speculated that under HTS, accumulation of Trp in *Z. rouxii* may require further synthesis via free Ser, resulting in the absence of Ser in the free amino acids.

### 3.5. RT-qPCR Analysis of Proteolysis Genes under Two Conditions

The dynamic adjustment of AA-P and AA-F in *Z. rouxii* under HTS not only indicated the response mechanism of amino acid biosynthesis but also reflected the ability of proteostasis. Proteostasis requires yeast to eliminate the production of damaged proteins by stress through proteolysis. Romero et al. [21] thought that yeast proteolysis was mediated by the ubiquitin-proteasome system. On the basis of transcriptome results and references, eight genes for ubiquitin-proteasome system in *Z. rouxii* were screened (Table 2) and verified by RT-qPCR analysis. Only *PRE4* (Proteinase YscE), *PSH1* (E3 ubiquitin ligase), and *CSE4* (Histone H3-like centromeric protein) were upregulated under control conditions (Figure 6A). In contrast, these eight genes were upregulated under HTS (Figure 6B), and the upregulation of *PRE4*, *PSH1*, and *CSE4* was more than as high as that of the control. The results confirmed that proteolysis still occurred in *Z. rouxii* at normal temperatures, but the expression of proteolysis genes was further promoted under HTS. Further analysis (Figure 6) showed that the most upregulated proteolysis gene under HTS was *PSH1* (13.23 ± 1.44 fold, *p* < 0.0001), while the most upregulated proteolysis gene in the control was *CSE4* (4.51 ± 0.96 fold, *p* < 0.0001). As can be seen from Table 2, both *PSH1* and *CSE4* correspond to histone proteolysis. It was speculated that histone proteolysis played an important role in maintaining proteostasis in *Z. rouxii*.

## 4. Discussion

### 4.1. The Importance of the Lag Phase

Most *S. cerevisiae* does not survive in this high-glucose (300 g/L glucose) environment, but we found that *Z. rouxii* was an optimal glucose environment at high temperatures. And the initiation response during the lag phase was very similar to that of the control, so we wanted to focus on the response during the lag phase to explain the positive effect of its thermic and osmotic stress on the growth of *Z. rouxii*. The set of physiological and metabolic changes occurring immediately after inoculation and during the lag phase is thought to be of vital importance for optimal offset of fermentation [30]. By measuring the compared viable cell count (Figure 1), remaining nitrogen source content (Figure 3A), and total protein content (Figure 2A), the exact point in time at which the two conditions were comparable under the lag phase was chosen. We will validate this program under other stress conditions or with other yeasts to test its applicability.

In this study, we showed that AA-P (Figure 2B–D) also significantly increased the number of lag phases under HTS. Other authors have also observed this phenomenon in some microbes. Alharbi [31] found that during the lag phase, *L. monocytogenes* ScottA rapidly altered its proteome (within 0.5 h), partially rebounding from the aged state but not completely reattaining the original exponential phase state; this was evident at all time points. Singh et al. [32] monitored the Raman spectra of individual *S. cerevisiae* cells for up to 3 h, during which the cells synthesize new proteins and lipids. Lag phase-induced proteins played a role in carbohydrate metabolism, protein synthesis, and amino acid metabolism. Compared to the control, not only did the amino acid species in AA-P differ significantly, but the species in AA-F (Figure 3) also changed significantly. This difference indicated that under sustained heat stress, *Z. rouxii* was capable of flexibly adjusting its protein and amino acid levels to adapt to the environment and prepare for cellular growth and proliferation during the log phase.

### 4.2. The Significance of Proteolysis under Stress

Shibata et al. [33] found that the duration of the lag phase was closely related to damage recovery mechanisms, with the contribution of cell damage to the total growth delay being 78%. Proteolysis is an important metabolic pathway for proteostasis to maintain cellular homeostasis, dealing mainly with proteins that are damaged in response to stress [34]. Therefore, studying the efficiency of proteolysis is important for understanding the physiological response to the lag phase under stressful conditions. The two main proteolysis systems present in the cell are the ubiquitin-proteasome system and the autophagy mechanism, which are coupled to each other [35]. The ubiquitin-proteasome system is a major degradation system for short-lived proteins. Prompt removal of these proteins is critical to the precise and timely regulation of intracellular signaling involved in multiple cellular processes, including cell proliferation and cell death [36].

The large number of genes makes it very difficult to study them as a whole. Therefore, many researchers have conducted studies on specific enzymes in this system. Hickey et al. [37] found E3s (Doa10, San1, and Ubr1) in *S. erevisiae* that function to remove proteins carrying two types of degradation signaling in response to quality control. Therefore, we aimed to more fully demonstrate the role of protein hydrolysis genes under HTS. In this paper, we combined transcriptomic data to select for validation the ubiquitin-activating enzyme E1 (*UBA3*), ubiquitin-binding enzyme E2 (*UBC7*), ubiquitin-protein ligase E3 (*PSH1*), and proteasome (*PRE4*), as well as other related enzymes (*CDC20, CDC23, CSE4,* and *RUB1*). This was only the transcript level performance of a small number of representative proteolysis genes but showed an absolute upregulation at HTS (Figure 6). This suggested that the sustained protein damage in *Z. rouxii* caused by HTS was high and that the cells themselves require sustained proteolysis to resist the stress response or they will fail to proliferate.

We found that *PSH1* was significantly highly expressed in the ubiquitination-mediated protein degradation of *Z. rouxii*, particularly under HTS. The presumed role of *PSH1* in regulating the ubiquitination of key growth proteins in yeast coincides with the ideas of other researchers. Cheng et al. [38] found that the centromere CenH3-labeled chromosomal locus, which is essential for genome stability and cell division, can undergo ubiquitination through selective recognition of yeast CenH3 (Cse4) by Psh1 in *S. cerevisiae*. This process controls the proteolysis of Cse4 at the cellular level. Histone chaperone Scm3 was found to impair Cse4 ubiquitination by eliminating Psh4-Cse1 binding, but whether Psh1 exerts the same effect in *Z. rouxii* deserves further investigation and will be the focus of subsequent attention by our team.

### 4.3. Role of ARO8 Expression in Z. rouxii

Aro8 was involved in several otherwise unrelated metabolic pathways. One of them, Aro8, is an aromatic amino acid transaminase and is involved in the biosynthesis of aromatic amino acids under many conditions. It has broad substrate specificity for aromatic amino acids, Leu, Met, and Glu as amino donors and the corresponding 2-oxo acids as amino acceptors [39]. The second isoform can be expressed as an alpha-aminoadipate transaminase (AAA-AT), which is involved in the biosynthesis of Lys and catalyzes the amination of 2-oxoadipate to α-aminoadipate [40]. According to the metabolic pathway descriptions in the KEGG database (Figure 5A–C), the functional description of Aro8 was present in *Z. rouxii*.

According to the GEO profile database (https://www.ncbi.nlm.nih.gov/geoprofiles/18438862) *ARO8* in *S. cerevisiae* was influenced by the time course of the heat shock stress response. Our results also indicate that the expression of *ARO8* in *Z. rouxii* was also downregulated at high temperatures during the lag phase. Furthermore, the inhibition of *ARO8* may indicate that the biosynthesis of Tyr, Lys, and Phe was affected (Figure 6). To confirm the effects of Tyr, Lys, and Phe on *Z. rouxii* at the physiological level. HTS conditions (40 °C, 300 g/L glucose, and 4.0 g/L L-proline) were used as the background for the experiments. We found that Phe, Lys, and Tyr (Figure 7) further promoted the growth of *Z. rouxii* under HTS. However, according to the results (Figure 3), free Phe was still present. This indicated that the *ARO8* gene preferentially produces aromatic amino acid transaminase when its expression is suppressed. This gene encodes an aromatic amino acid transaminase enzyme and was able to higher binding affinity phenylpyruvate rather than 4-hydroxyphenylpyruvate under HTS. For *ARO8*, functionality goes far beyond that.. Guo et al. [41] showed that the aminotransferase Aro8 from *S. cerevisiae* has broad substrate ranges and can catalyze the conversion of three kinds of aromatic amino acids (Trp, Tyr, or Phe) to the corresponding indole-3-acetic acid, 4-hydroxyphenylacetic acid and phenylacetic acid. Ohashi and Chaleckis [42] found Trp degradation deficient mutant Aro8, Aro9 *S. cerevisiae* cells, in which Trp accumulation was confirmed, were sensitive to several CW/M stresses. According to (Figure 3C), Trp content in AA-F gradually dominates over time course under HTS. This suggests that *Z. rouxii*, like *S. cerevisiae*, through Trp accumulation participates in the cell wall integrity (CWI) pathway and reduces the cell wall or membrane (CW/M) stress tolerance.

The specific function of Phe, the focus of observation in this paper, is not yet clear. According to other researchers, Phe was found to affect the ability of *S. cerevisiae* to produce higher alcohols. Sun et al. [43], by metabolite analysis and transcriptome sequencing, showed that the metabolic pathways of branched-chain amino acids, pyruvate, Phe, and Pro were the decisive factors that affected the formation of higher alcohols. We will follow up with an analysis of the metabolites of *Z. rouxii* under HTS to verify whether they would have the same effect in *Z. rouxii*. The translation process of mRNA is affected by stress interference in a number of ways [44,45], but the efficiency of the supply of amino acids, the precursors of protein synthesis, is ignored. Our study has shown that HTS significantly alters amino acid anabolic metabolism during the acclimation period in *Z. rouxii*. Detailed studies are needed to understand how amino acid metabolism and protein translation are coordinated and bridged under stress.

## 5. Conclusions

This is the first study of the response of lag phase proteins to HTS in *Z. rouxii*. We observed that AA-P and AA-F changed rapidly, suggesting an intense response of the proteins to stressful environments. Genes in the ubiquitin-proteasome system were upregulated, indicating that proteolysis plays an important role in stress. In conclusion, we found that proteostasis plays an important role in *Z. rouxii* in the face of HTS. Starting from the lag phase, proteostasis allows *Z. rouxii* to adapt to its environment to continue proliferating under prolonged HTS. We also found that *ARO8* preference encoding Tyr, Lys, and Phe further promoted *Z. rouxii* growth under HTS. Understanding the stress response of yeast may help to optimize fermentation processes and improve product quality. Further studies are needed to explore the proteolysis and expression patterns in other budding yeasts grown under HTS, which may reveal new mechanisms of stress adaptation and have practical applications in the industry.

## Figures and Tables

**Figure 1 jof-10-00048-f001:**
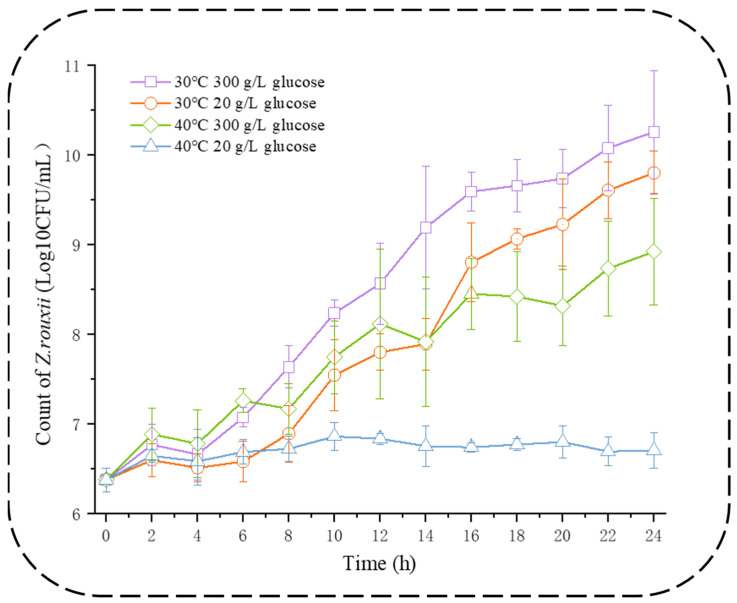
Growth curve analysis of *Z. rouxii* under two temperatures. Data are presented as the mean ± SEM from three independent experiments.

**Figure 2 jof-10-00048-f002:**
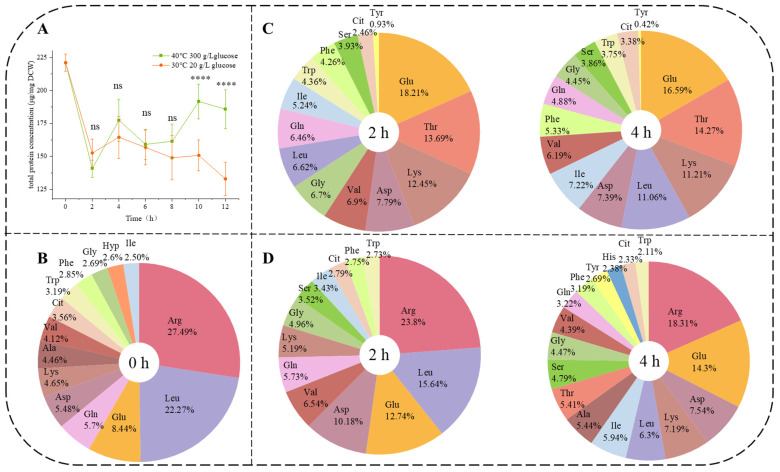
Trend of total protein content and distribution of amino acid (AA-P) percentage of lag phase. (**A**) First 12 h total protein concentration in the *Z. rouxii* cells, (**B**) initial amino acid composition of the protein (0 h), (**C**) proteolysis amino acid composition under control during the lag phase (2–4 h), (**D**) proteolysis amino acid composition under HTS during the lag phase (2–4 h). Data (**A**) were presented as the mean  ±  SEM from three independent experiments. Different symbols (ns-****) represent significant differences between the two conditions within the same study hours (*p* < 0.05). Data (**B**–**D**) were presented as the mean from three independent experiments.

**Figure 3 jof-10-00048-f003:**
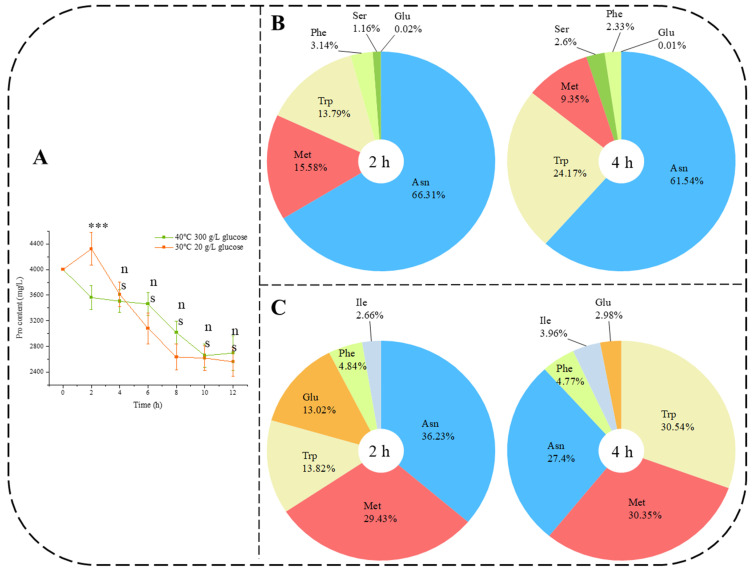
Consumption of added amino acids and the percentage of free amino acid (AA-F) during lag phase. (**A**) The remaining Pro concentration in the synthetic medium for the first 12 h, (**B**) free amino acid composition (except Pro) under control during the lag phase (2 h–4 h), (**C**) free amino acid composition under HTS during the lag phase (2 h–4 h). Data (**A**) were presented as the mean ± SEM from three independent experiments. Different symbols (ns-***) represent significant differences between the two conditions within the same study hours (*p* < 0.05). Data (**B**,**C**) were presented as the mean from three independent experiments.

**Figure 4 jof-10-00048-f004:**
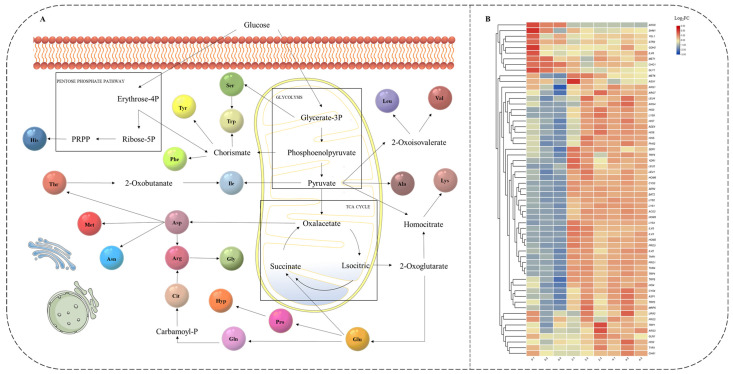
Amino acid biosynthesis pathway map and its gene transcription level expression results of lag phase under HTS. (**A**) Biosynthesis of amino acids—*Zygosaccharomyces rouxii* from KEGG (zro01230). (**B**) Heatmap of transcriptome differentially expression results of under HTS during the lag phase (0 h–2 h–4 h) amino acid biosynthesis pathways genes corresponding to KEGG—*Z. rouxii* (zro01230). RSEM: TPM, Log2FC > 2 and FDR-*p*-value < 0.05. Data were presented as three independent experiments.

**Figure 5 jof-10-00048-f005:**
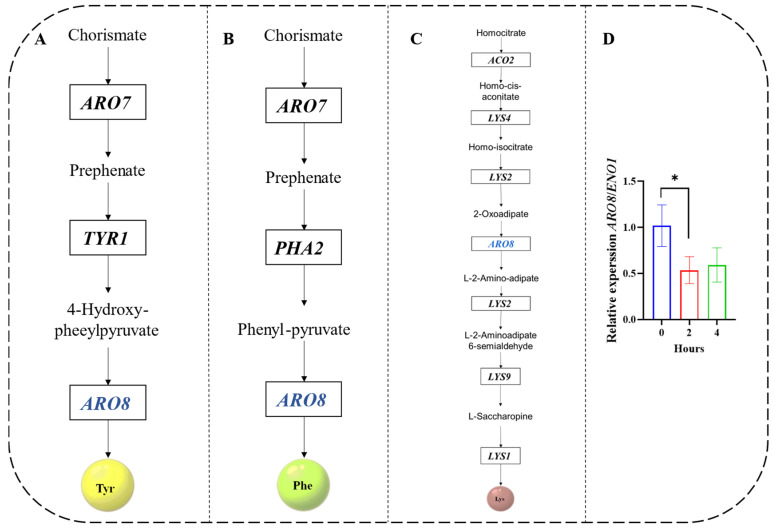
Key genes for Phe, Tyr, and Lys biosynthesis pathways according to KEGG—*Z. rouxii* and validation of RT-qPCR levels of amino acids biosynthesis gene downregulated by transcriptional under HTS, (**A**) Tyr biosynthesis pathway (zro_M00025), (**B**) Phe biosynthesis pathway (zro_M00024), (**C**) Lys biosynthesis pathway (zro_M00030), (**D**) *ARO8* RT-qPCR analysis under high-temperature stress during the lag phase (0 h–2 h–4 h). Data (**D**) were presented as the mean  ±  SEM from three independent experiments. One-way ANOVA with 0 h as reference. Different symbols (*) represent the significant differences between hours within the same study conditions (*p* < 0.05).

**Figure 6 jof-10-00048-f006:**
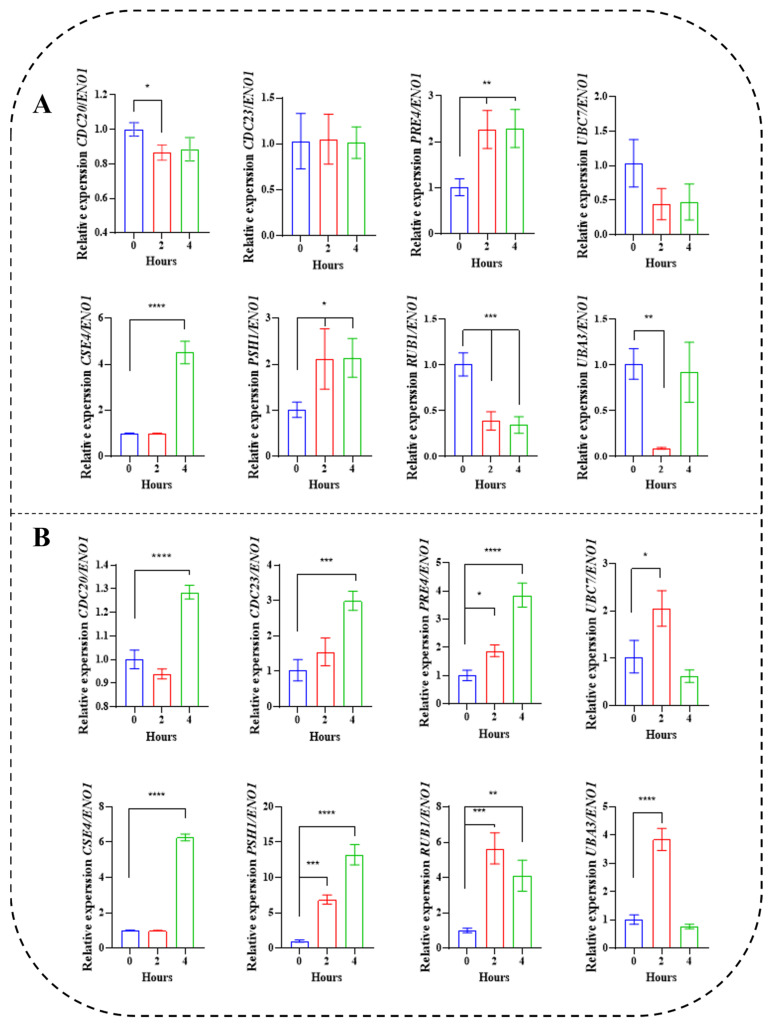
Time course analysis inside 2 conditions 8 proteolysis genes RT-qPCR results: (**A**) control during the lag phase (0 h–2 h–4 h); (**B**) HTS during the lag phase (0 h–2 h–4 h). Data were presented as the mean  ±  SEM from three independent experiments. One-way ANOVA with 0 h as reference. Different symbols (*, **, *** and ****) represent significant differences between hours within the same study conditions (*p <* 0.05).

**Figure 7 jof-10-00048-f007:**
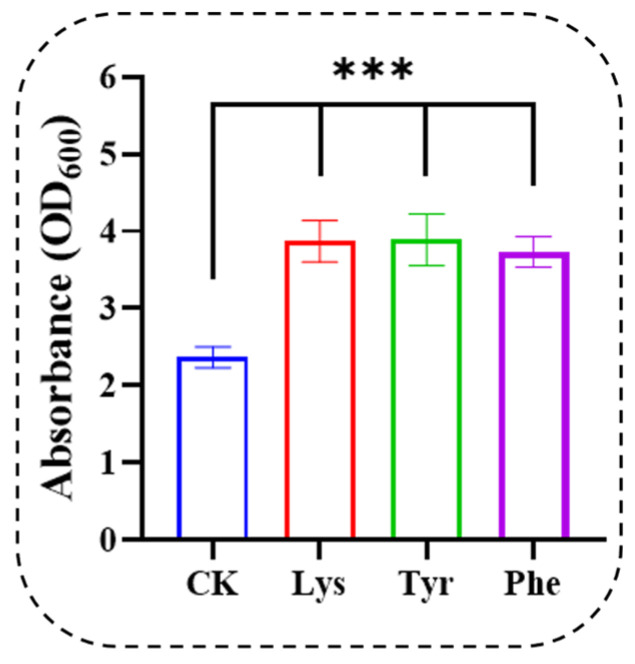
Lys, Tyr, or Phe was added to the 40 °C 300 g/L glucose condition to determine yeast cell density (OD_600_) at 120 h. CK stands for 40 °C 300 g/L glucose condition. Data were presented as the mean ± SEM from three independent experiments. One-way ANOVA with CK as reference. Different symbols (***) represent the significant differences between amino acids within the same study conditions (*p* < 0.05).

**Table 1 jof-10-00048-t001:** Specific primers of the reference gene and related genes for real-time qPCR analysis.

Gene	Primers	Sequence (5′→3′)
*CDC20*	F	CTACATGGGCAGAGCACACA
	R	ACAGCAGCAGTATGCGTCTT
*CDC23*	F	TTTGGTTTGGGCCAGGCTTA
	R	TCCGACGATCCCAAGGTTTC
*PRE4*	F	CACACATGGCAAACCCACTG
	R	CGCCTCTTCAGCAACCTGTA
*UBC7*	F	GCTGGTCCCAAATCGGAGAA
	R	CACCATCAGCGTATGGCGTA
*CSE4*	F	AGCTTAAAGCGCGTCGAAAA
	R	CCTGTAACGCCATAATCGCC
*PSH1*	F	GCGTCTACCAATCCGTTTGC
	R	TATGTTCTCGCATGGGCTCC
*RUB1*	F	GAAGACACTGACTGGGAAGGAG
	R	TGTTGAGATGGTGGAATCCCT
*UBA3*	F	CTCAAGTGGCAGCCCAGTAT
	R	TAAAAGCTCGGGGGCAAAGT
*ARO8*	F	AACGCTACGCTATCCAGGTG
	R	CGATGGCACAGTCACGTTTC

**Table 2 jof-10-00048-t002:** Functional annotation of proteolysis genes.

Gene	Enzyme Name	Function	References
*CDC20*	A highly conserved activator of the anaphase-promoting complex (APC)	promoting cell cycle-regulated ubiquitination and proteolysis of several critical cell cycle-regulatory targets including securin and mitotic cyclins	[22]
*CDC23*	Anaphase-promoting complex subunit	shown to play a role in cyclin proteolysis	[23]
*PRE4*	Proteinase YscE	in cleavage of peptide bonds after acidic amino acids	[24]
*UBC7*	Ubiquitin-conjugating enzyme E2	ubiquitination and proteolysis of D2 (type 2 mono deiodinase)	[25]
*CSE4*	Histone H3-like centromeric protein	the C-terminus of Cse4 contributes to CenH3 (centromeric histone H3) proteolysis	[26]
*PSH1*	an E3 ubiquitin ligase	contribute to CenH3 proteolysis	[27]
*RUB1*	A ubiquitin-like protein displaying a 53% amino acid identity to ubiquitin	the RUB1 conjugation pathway is functionally affiliated with the ubiquitin-proteasome system and may play a regulatory role	[28]
*UBA3*	E1 ubiquitin-activating enzyme	C-terminal domains of the E1 ubiquitin-activating enzyme	[29]

## Data Availability

New high throughput sequencing datasets (RNA-seq, ChIP-Seq, degradome analysis, etc.) must be deposited in either the NCBI’s Sequence Read Archive (SRA PRJNA975303).

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
