# Peer review of "The Protein Response of Salt-Tolerant Zygosaccharomyces rouxii to High-Temperature Stress during the Lag Phase"

_jof, 2024, doi:10.3390/jof10010048_

Round 1

Reviewer 1 Report

Comments and Suggestions for Authors

In this manuscript, Hu and co-authors present an interesting work set to study the proteostasis in the salt-tolerant Zygosaccharomyces rouxii exposed to high-temperature stress during the lag phase, with a special focus on the regulation of proteolysis genes under heat stress conditios. A comprehensive transcriptome sequencing was done to reveal the potential genes involved in proteostasis associated with heat stress. The study presents clear and sound data organized in a coherent and logical manuscript.

There are some issues that the authors need to address before the manuscript can be considered for publication.

-        - The authors should explain the rationale behind growing the Z. rouxii strain at high temperature under high glucose conditions (i. e. 300 g/L). This is 15 times higher than the normal 20 g/L glucose, a concentration at which most Saccharomyces cerevisiae strains do not survive.

-       -  Line 105: please specify the type of beads used to break the cells.

-        - Line 123: please define DEPC.

-        - Figure 2 is difficult to understand due to small fonts, which become blurry when attempting to enlarge the page. The same applies to Figure 4.

Comments on the Quality of English Language

Minor editing is of English is necessary. Also, please italicize the binominal species names all throughout the manuscript.

Author Response

Thanks to the experts for their valuable suggestions, all changes have been highlighted in red font in the article, thanks again for your valuable suggestions. My detailed answers to the questions raised are set out below:

Question 1. The authors should explain the rationale behind growing the Z. rouxii strain at high temperature under high glucose conditions (i. e. 300 g/L). This is 15 times higher than the normal 20 g/L glucose, a concentration at which most Saccharomyces cerevisiae strains do not survive.

Respond: Thank you for your valuable suggestions. This is indeed something that needs to be mentioned in this manuscript, the results of the discussion have been added in 351-355. “Most S. cerevisiae does not survive in this high-glucose (300 g/L glucose) environment, but We found Z. rouxii was an optimal glucose environment at high temperatures. And the initiation response during the lag phase was very similar to that of the control, so we wanted to focus on the response during the lag phase to explain the positive effect of its thermic and osmotic stress on the growth of Z. rouxii.”

Question 2. Line 105: please specify the type of beads used to break the cells.

Respond: Thank you for your comments, which have been corrected in the text. Corrected to” 0.5 mm glass beads”

Question 3. Line 123: please define DEPC.

Respond: Thank you for your comments, which have been corrected in the text. Corrected to “DEPC (diethyl pyrocarbonate)”

Question 4. Figure 2 is difficult to understand due to small fonts, which become blurry when attempting to enlarge the page. The same applies to Figure 4.

Respond: Thank you for your comments. We have reformatted the images, it is possible that the lack of clarity is due to saving issues, but we have lossless images in PPT format and will make further corrections if further adjustments are needed.

Comments on the Quality of English Language

Question 5. Minor editing is of English is necessary. Also, please italicize the binominal species names all throughout the manuscript.

Respond: Thank you for your comments. We have corrected the article for strain names and other issues, and checked for English writing problems.

Thank you again for the valuable comments of the experts, has been revised, looking forward to the experts for criticism and correction. Please see the attachment for the revised manuscript.

Reviewer 2 Report

Comments and Suggestions for Authors

This work describes that Zygosaccharomyces rouxii was cultured using a synthetic medium containing only proline as a nitrogen source to know the protein level at high-temperature stress during the lag phase. The authors claim that differential expression analysis of the amino acid biosynthesis-related genes in the transcriptome showed that most genes were upregulated under high-temperature stress. They also declare that high-temperature stress significantly increased the upregulation of proteolysis genes. Finally, they concluded that Z. rouxii adapts to prolonged high-temperature stress by altering its basal protein composition, which was related to the regulation of proteolysis and the biosynthesis of amino acids.

The work is well-designed, and the conclusion seems well-supported. Nonetheless, the following observations/questions arise.

  1. Why did authors select the biosynthesis of amino acids and the ubiquitin-proteasome system as the pathways for studying the yeast's stress response? Please explain.
  2. The culture at high glucose concentration (300 g/L) can be considered under thermic and osmotic stress. Please discuss it.
  3. Why did the authors replace Pro with Lys, Tyr, and Phe?

  1. The proline content of 4 g/L in the culture medium means a nitrogen content of 0.488 g N/L. Then, the medium with a glucose content of 20 g/L had a C/N ratio of 16.39 g C/g N. It was a medium with good nitrogen content to support good yeast growth. Nevertheless, the medium with a glucose content of 300 g/L had a C/N ratio of 245.9 g C/ g N, with poor nitrogen content. How could these differences in the nutritional quality of the medium affect the results of this study? Please try to discuss this point.

  1. The glucose consumption was not measured, or it is not included in the manuscript. Nevertheless, it could be significant because substantial differences in maintenance energy could be found between fermentation kinetics shown in Fig. 1. If the authors have these results, please include and discuss them. If not, please discuss its potential importance.

SPECIFIC COMMENTS

Line 43. Please write Saccharomyces cerevisiae in italics, also on Line 59.

Line 75-76. Please provide the glycerol concentration used for the strain conservation at -80°C.

Line 173-175. This paragraph seems unnecessary. Please consider deleting it.

Line 294 (Fig. 1). Please write "g/L" rather than "g/l"

Line 297 (Fig 2). The Fig. 2A need a better definition. Please correct.

Line 305 (Fig. 3). The Fig. 3A need a better definition. Please correct.

Line 342-350. The content of these lines seems more suitable for the introductory section of the manuscript. Please consider removing it from the discussion and integrating it into the introduction.

Comments on the Quality of English Language

The English language needs to be reviewed by a native speaker. The manuscript is generally understandable, but it contains many spelling, grammar and punctuation mistakes. For example, the sentence "Differential expression analysis of the amino acid biosynthesis-related genes in the transcriptome showed that most of the genes were upregulated under HTS, excluded ARO8 which was consistently repressed during the lag phase." would be changed to the sentence "Differential expression analysis of the amino acid biosynthesis-related genes in the transcriptome showed that most genes were upregulated under HTS, excluding ARO8, which was consistently repressed during the lag phase."

Additionally, some sentences could be shorter, and it would be better to split them into two sentences for better clarity. For example, the sentence (Lines 13-17) "Within the first two h, the total intracellular protein concentration was significantly decreased from 220.99 ± 6.58 μg/mg DCW to 152.63 ± 10.49 μg/mg DCW, and the results from the analysis of the amino acid composition of the total protein through vacuum proteolysis technology and HPLC showed that new amino acids (Thr, Tyr, Ser, and His) were added to newborn protein over time course during the lag phase under HTS." could be changed to the two sentences "Within the first two h, the total intracellular protein concentration was significantly decreased from 220.99 ± 6.58 μg/mg DCW to 152.63 ± 10.49 μg/mg DCW. The analysis of the amino acid composition of the total protein through vacuum proteolysis technology and HPLC showed that new amino acids (Thr, Tyr, Ser, and His) were added to newborn protein overtime during the lag phase under HTS."

Author Response

Thanks to the experts for their valuable suggestions, all changes have been highlighted in red font in the article, thanks again for your valuable suggestions.

Question 1. Why did authors select the biosynthesis of amino acids and the ubiquitin-proteasome system as the pathways for studying the yeast's stress response? Please explain.

Respond: Thank you for your valuable suggestions. Because we found that the change in carbon source in Z. rouxii in response to high temperature stress did not give a substantial increase in the sugar consumption ratio, we then investigated the change in its nitrogen source and found an increase in the number of free and fixed amino acid species. Since this medium has only proline as a source, it is known from literature references that amino acid species can only be introduced by glucose for amino acid biosynthesis without external addition of new amino acids. Later its total protein content was measured and found to be significantly changed, this change under stress requires the ubiquitin-proteasome system to form new fixed amino acid combinations after enzymatic degradation to form proteins.

Question 2. The culture at high glucose concentration (300 g/L) can be considered under thermic and osmotic stress. Please discuss it.

Respond: Thank you for your valuable suggestions, the results of the discussion have been added in 351-354 “Most S. cerevisiae does not survive in this high-glucose (300 g/L glucose) environment, but We found Z. rouxii was an optimal glucose environment at high temperatures. And the initiation response during the lag phase was very similar to that of the control, so we wanted to focus on the response during the lag phase to explain the positive effect of its thermic and osmotic stress on the growth of Z. rouxii.”

Question 3. Why did the authors replace Pro with Lys, Tyr, and Phe?

Respond: Thank you for your valuable suggestions. Because we pre-laboratory performed amino acid screening experiments on Z. rouxii media, based on 30°C, and validated at 40°C and found both to be optimal for proline. Tyr and Phe were able to grow but were clearly undergrown at 40°C because we chose to perform the experiments in synthetic medium with only proline added. The figure shows the results of yeast growth under the two conditions in the manuscript with OD600=0.2 as the amino acid replacement inoculum for the medium.

Question 4. The proline content of 4 g/L in the culture medium means a nitrogen content of 0.488 g N/L. Then, the medium with a glucose content of 20 g/L had a C/N ratio of 16.39 g C/g N. It was a medium with good nitrogen content to support good yeast growth. Nevertheless, the medium with a glucose content of 300 g/L had a C/N ratio of 245.9 g C/ g N, with poor nitrogen content. How could these differences in the nutritional quality of the medium affect the results of this study? Please try to discuss this point.

Respond: Thank you for your valuable suggestions, the results of the discussion have been added in 187-195 “It is generally accepted that C/N ratio of 16.39 g C/g N (20 g/L glucose) was a medium with good nitrogen content to support good yeast growth, so we chose 30°C 20 g/L glucose was selected as the control. Meanwhile, we observed that Z. rouxii grows better in a medium with 300 g/L glucose at a C/N ratio of 245.9 g C/g N, and this is especially evident at high temperatures. However, in most cases, we believe that a C/N ratio of 245.9 g C/g N with a worse nitrogen source should be detrimental to growth, so we further compared the fixed and free amino acids of these two conditions to understand the response mechanism of Z. rouxii under high temperature and high osmotic stress.”

Question 5. The glucose consumption was not measured, or it is not included in the manuscript. Nevertheless, it could be significant because substantial differences in maintenance energy could be found between fermentation kinetics shown in Fig. 1. If the authors have these results, please include and discuss them. If not, please discuss its potential importance.

Respond: Thank you for your valuable suggestions. We have measured its sugar consumption, and although there seems to be a significant difference in quantity, the glucose consumption ratio has not increased dramatically, so we thought that it is not the main source of energy, but instead it is possible that this high osmosis from the outside world protects it against high temperature stress. This will be one of the directions of our subsequent research, which will be devoted to it and a new paper will be written separately. The graph shows the sugar consumption rates for the two experimental conditions of the article.

SPECIFIC COMMENTS

Question 6. Line 43. Please write Saccharomyces cerevisiae in italics, also on Line 59.

Respond: Thank you for your comments, which have been corrected in the text.

Question 7. Line 75-76. Please provide the glycerol concentration used for the strain conservation at -80°C.

Respond: Thank you for your comments, which have been corrected in the text. Add “(strain/glycerol = 1/1, V/V))”

Question 8. Line 173-175. This paragraph seems unnecessary. Please consider deleting it. Respond: Thank you for your comments. We have now deleted.

Question 9. Line 294 (Fig. 1). Please write "g/L" rather than "g/l".

Respond: Thank you for your comments, which have been corrected in the text.

Question 10. Line 297 (Fig 2). The Fig. 2A need a better definition. Please correct.

Respond: Thank you for your comments, which have been corrected in the text. Corrected to “Trend of total protein content and distribution of amino acid (AA-P) percentage of lag phase”

Question 11. Line 305 (Fig. 3). The Fig. 3A need a better definition. Please correct.

Respond: Thank you for your comments, which have been corrected in the text. Corrected to “Consumption of added amino acids and the percentage of free amino acid (AA-F) during lag phase”

Question 12. Line 342-350. The content of these lines seems more suitable for the introductory section of the manuscript. Please consider removing it from the discussion and integrating it into the introduction.

Respond: Thank you for your comments, which have been corrected in the text. Line 342-350 has been added to the preamble Line 67-75. “Meanwhile, lag is a dynamic, organized, adaptive, and evolvable process that protects bacteria from threats, promotes reproductive fitness, and is broadly relevant to the study of bacterial evolution, host-pathogen interactions, antibiotic tolerance, environmental biology, molecular microbiology, and food safety (Bertrand, 2019). However, this was still a small number of studies, with the majority focusing on changes in the stress response of log phase yeast. Because of the short duration of the lag phase and the unclear composition of the medium, it was difficult to obtain valid data. Therefore, in this paper, we created a synthetic medium in which yeast cell was grown in minimal nutrition condition.”

Comments on the Quality of English Language

Question 13. The English language needs to be reviewed by a native speaker. The manuscript is generally understandable, but it contains many spelling, grammar and punctuation mistakes. For example, the sentence "Differential expression analysis of the amino acid biosynthesis-related genes in the transcriptome showed that most of the genes were upregulated under HTS, excluded ARO8 which was consistently repressed during the lag phase." would be changed to the sentence "Differential expression analysis of the amino acid biosynthesis-related genes in the transcriptome showed that most genes were upregulated under HTS, excluding ARO8, which was consistently repressed during the lag phase."

Respond: Thank you for your comments, which have been replaced in the text with “Differential expression analysis of the amino acid biosynthesis-related genes in the transcriptome showed that most genes were upregulated under HTS, excluding ARO8, which was consistently repressed during the lag phase."

Question 14. Additionally, some sentences could be shorter, and it would be better to split them into two sentences for better clarity. For example, the sentence (Lines 13-17) "Within the first two h, the total intracellular protein concentration was significantly decreased from 220.99 ± 6.58 μg/mg DCW to 152.63 ± 10.49 μg/mg DCW, and the results from the analysis of the amino acid composition of the total protein through vacuum proteolysis technology and HPLC showed that new amino acids (Thr, Tyr, Ser, and His) were added to newborn protein over time course during the lag phase under HTS." could be changed to the two sentences "Within the first two h, the total intracellular protein concentration was significantly decreased from 220.99 ± 6.58 μg/mg DCW to 152.63 ± 10.49 μg/mg DCW. The analysis of the amino acid composition of the total protein through vacuum proteolysis technology and HPLC showed that new amino acids (Thr, Tyr, Ser, and His) were added to newborn protein overtime during the lag phase under HTS.

Respond: Thank you for your comments, which have been replaced in the text with” "Within the first two h, the total intracellular protein concentration was significantly decreased from 220.99 ± 6.58 μg/mg DCW to 152.63 ± 10.49 μg/mg DCW. The analysis of the amino acid composition of the total protein through vacuum proteolysis technology and HPLC showed that new amino acids (Thr, Tyr, Ser, and His) were added to newborn protein overtime during the lag phase under HTS."

Thank you again for the valuable comments of the experts, has been revised, looking forward to the experts for criticism and correction. Please see the attachment for the revised manuscript.

Round 2

Reviewer 1 Report

Comments and Suggestions for Authors

The authors responded to all reviewer's concerns.